# Active Charge Balancing Strategy Using the State of Charge Estimation Technique for a PV-Battery Hybrid System

**Md Ohirul Qays [1], Yonis Buswig [1], Md Liton Hossain [2],\* and Ahmed Abu-Siada [2]**

[1] Department of Electrical and Electronic Engineering, Faculty of Engineering, Universiti Malaysia Sarawak, Kota Samarahan 94300, Sarawak, Malaysia; 18020101@siswa.unimas.my (M.O.Q.); byonis@unimas.my (Y.B.)

[2] School of Electrical Engineering, Computing and Mathematical Sciences, Curtin University, Perth 6102, Australia; a.abu-siada@curtin.edu.au

\* Correspondence: mdliton.hossain@postgrad.curtin.edu.au

**Abstract:** Charging a group of series-connected batteries of a PV-battery hybrid system exhibits an imbalance issue. Such imbalance has severe consequences on the battery activation function and the maintenance cost of the entire system. Therefore, this paper proposes an active battery balancing technique for a PV-battery integrated system to improve its performance and lifespan. Battery state of charge (SOC) estimation based on the backpropagation neural network (BPNN) technique is utilized to check the charge condition of the storage system. The developed battery management system (BMS) receives the SOC estimation of the individual batteries and issues control signal to the DC/DC Buck-boost converter to balance the charge status of the connected group of batteries. Simulation and experimental results using MATLAB-ATMega2560 interfacing system reveal the effectiveness of the proposed approach.

**Keywords:** active battery balancing; backpropagation neural network; DC/DC Buck-boost converter; PV-battery integrated system; state of charge estimation

## 1. Introduction

In the last few decades, photovoltaics (PV) have been broadly used as a cost effective and reliable renewable energy source with the aim of reducing the reliance on fossil fuel used in conventional thermal generation [1]. In [2], a detailed PV model is investigated using an artificial neural network (ANN). However, adopting such model in practical PV systems increases the implementation time and complexity when compared to classical models. A maximum power point tracking (MPPT) system is essential for PV systems to yield the maximum accessible power irrespective the irregular solar irradiance and atmospheric condition. A combination of Adaptive neuro-fuzzy inference system (ANFIS) and hill climbing (HC) MPPT technique is presented in [3]. Reported results outperformed the tracking precision of the conventional MPPT technique. To increase the profit of the PV system during partial shading events, an adaptive perturb and observation (P&O) MPPT algorithm is proposed [4]. However the proposed algorithm is only valid for monocrystalline and polycrystalline-based PV panels. Ramp and step alterations of solar radioactivity was used to test the developed P&O MPPT technique. Results show that the proposed technique is of quicker tracking speed than the conventional method.

During cloudy days and night-time, battery energy storage systems (BESS) play an important role for providing off-grid power [5–7]. In [8], a 1.5 kW PV-battery integrated system is simulated using HOMER software to assess the battery lifetime. Simulation was conducted by keeping the battery depth of discharge (DOD) within 50~80%. The research reveals that the greenhouse gas (GHG) effect of the

atmosphere can be reduced by improving the battery lifespan through a proper charging/discharging control system during the off-grid condition. The SOC of a 48 V valve-regulated lead-acid battery (VRLA) is estimated in [9] through the sigma-point Kalman filter (SPKF) algorithm. The 2nd order Equivalent Circuit Model (ECM) was designed in MATLAB-PIC18F2550 interfacing system and tested via Unscented Kalman Filter (UKF) scheme.

Nevertheless, BESS comes across quite a few challenges, including battery to battery factorial delinquencies and cell performance. Figure 1 shows that a group of series connected batteries may encompass dissimilar levels of SOC. Thus, a minimum-labeled SOC battery is deep-discharged during the discharging period, while a maximum-labeled SOC battery will be over-charged in the charging period [10,11]. The least squares variable projection algorithm (LSVPA) for lithium-ion batteries (LIB) was employed in [12] in order to improve the noise immunity and unbiased model parameter identification. Reported results indicated a reduction of the root mean square error from 10.22 mV to 0.54 mV by using a MATLAB-Arbin BT2000 testing system. Wöhler-curve-based LIB battery functioning life is studied in [13] for economic assessment and sizing of battery systems in railway applications. The proposed method improved the battery lifetime compared to the semi-empirical method. In [14], a Luenberger observer is utilized along with instrumental variable estimation and bilinear principle combination to estimate the SOC for LIB. The proposed method resulted in 2.60% and 1.25% error in the case of hybrid pulse experiment and Federal Urban Dynamic Schedule conditions, respectively. In [15], four series-connected 12 V lead acid (LA) batteries were investigated for electric motorcycles. The circuit was designed using six MOSFET switches to improve the life-cycle of the battery bank. However, the proposed method still requires further experimentation for real-life application.

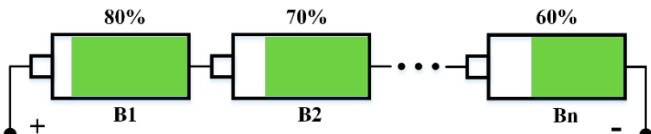

**Figure 1.** Different leveled SOC of individual batteries within a battery pack.

Two types of battery-cell balancing are classified as passive and active cell balancing techniques [16]. The passive balancing technique coverts the extra energy into heat through a resistor that is not approachable to surroundings whereas the active balancing technique allocates the energy among the batteries from which power losses are significantly reduced. A solar power-supported 48 V LA battery was explored for electric vehicles in [17]. Eight MOSFET switches and a National Instrument (NI) DAQ were implemented to control the balancing system for four individual 12 V LA batteries. Though the projected model increases the battery energy from 2.1% to 3.3% in every 13.2 km, the polarization and temperature effect of the battery was not considered. A real-time monitoring system for multiple LA batteries using an ATmega128a microcontroller-based BQ34Z110 board is proposed in [18]. WizFi210 is integrated with several types of sensors, such as humidity, temperature, and liquid level devices, to monitor the battery condition and calculate the SOC, which was not very accurate.

To the best of the authors' knowledge, the active charge equalization strategy based on SOC estimation approach using the BPNN scheme is a better solution to prolong the battery activation life for a PV-battery hybrid system, and no such research has been investigated previously. Hence, the foremost contribution of this paper is the presentation of an improved BMS through active cell balancing technique for a standalone PV-Battery integrated system. Additionally, an operative SOC estimation technique is proposed using a BPNN algorithm to prolong the functional life of a lead acid battery along with the Wöhler-curve-based aging model. The proposed model is validated through simulation and experimental analyses using a MATLAB-ATMega2560 interfacing system.

The rest of this paper is arranged as follows: Section 2 explains the modeling approach of the proposed model. Numerical simulation and experimental analysis are presented in Section 3. Key results and a discussion of the proposed methodology are deliberated in Section 4, while key conclusions are drawn in Section 5.

## 2. Modeling Approach

### 2.1. Photovoltaic (PV) Mathematical Modeling

The combination of series or parallel coupled solar cells regulate the generated power of PV panels. Principally, two types of PV cells are categorized as double-diode and single-diode modules [19,20]. Though double-diode modules exhibit high accuracy, low analytical speed is depicted due to the structural complexity. In contrast, single-diode modules are widely adopted in numerous power electronics application due to their high precision. Consequently, this paper focuses on a single-diode-based PV module which entails a current source ($I_{SC}$), diode $D$, and series-parallel resistors $R_s$, $R_{sh}$. From Figure 2, Kirchhoff's Current law formulates the output current ($I_{PV}$) as:

$$I_{PV} = n_P I_L - I_D - \frac{V_D}{R_{sh}} \tag{1}$$

where $I_D$ represents the diode current, $I_L$ is the light-generated current, number of parallel connected cells is $n_P$, shunt resistance is $R_{sh}$ and diode voltage is $V_D$. Diode current can be calculated from Equation (2):

$$I_D = n_P I_{OS} \left[ \exp\left( \frac{q(V + IR_s)}{n_s AKT} \right) - 1 \right] \tag{2}$$

where the reverse saturation current is denoted by $I_{OS}$, electron charge is $q$, output voltage is $V$, cell current is $I_{cell}$, series resistance is $R_s$, number of series connected cells is $n_s$, ideality factor is $A$, Boltzmann constant is $K$ and the cell temperature is $T$ in Kelvin.

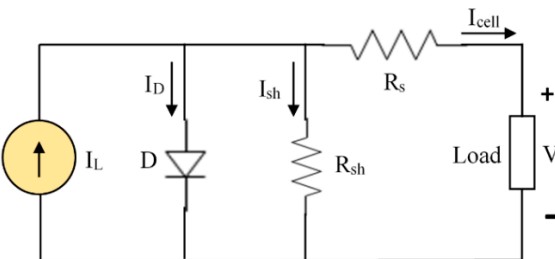

**Figure 2.** PV cell structure based on a single diode module.

The light-generated current ($I_L$) and reverse saturation current ($I_{OS}$) is affected by solar irradiance and atmospheric temperature and can be calculated from Equations (3) and (4):

$$I_L = \frac{\psi}{\psi_{sh}} [I_{Lref} + \mu I_{sc}(T - T_{ref})] \tag{3}$$

$$I_{OS} = D_f T^3 \exp\left( \frac{-qE}{AKT} \right) \tag{4}$$

where the existing irradiance and reference irradiance are represented by $\psi$ and $\psi_{sh}$. $I_{Lref}$ is the light current at reference condition, $T$ and $T_{ref}$ indicate the present and reference temperatures; respectively, $D_f$ is the diode diffusion factor and $E$ is the bandgap energy.

### 2.2. Maximum Power Point Tracking (MPPT)

The PV systems should be operated at maximum capability irrespective the disparities of environmental and loading circumstances. The maximum probable power can be taken out from the PV panel using a systematic prevailing appliance namely maximum power point tracking (MPPT). Generally, a DC-DC power converter is operated in conjunction with a PV array whose duty cycle is identified via the MPPT system. As shown in Figure 3, The determination of the MPPT is to adjust

the I–V functional point of PV modules to harvest the supreme accessible power from the electronic converter [21]. Several types of MPPT techniques comprising fuzzy logic based MPPT, artificial neural network (ANN), perturbation and observation (P&O), fractional open-circuit voltage (FOCV), incremental conductance (IC), look-up table, ripple correlation control (RCC) and one cycle control (OCC)-based MPPT can be adopted [22]. The authors of [19] reported that the fuzzy logic-based MPPT technique is the most truthful method compared to other approaches. A modified P&O-based MPPT technique is presented in [23] through MATLAB/SIMULINK software. Compared to the traditional P&O method, this algorithm was capable to recognize the voltage maximum power point (MPP) exactly. Nonetheless, experimental justification of this technique was recommended by the authors along with the temperature effect in future research.

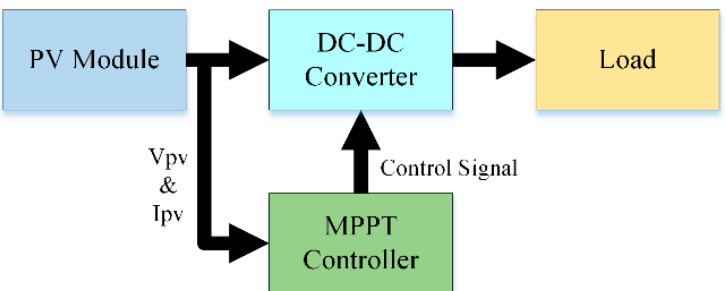

**Figure 3.** Block diagram of PV-MPPT governing system.

In order to improve the working prototype in real world application, a P&O-based MPPT system as shown in Figure 4 is presented in [24]. In this method, voltage and current collected from the PV panel are used to calculate the solar power at a time (t) which is compared with a former sample measured at time (t − 1). Voltage perturbation stays on the similar way, if the power alteration is positive. Negative power alteration (ΔP) decreases the voltage perturbation (ΔV) as can be seen in Table 1. Following this technique, MPP can be traced over the entire PV curve. Figure 4 summarizes the flowchart of the entire P&O algorithm.

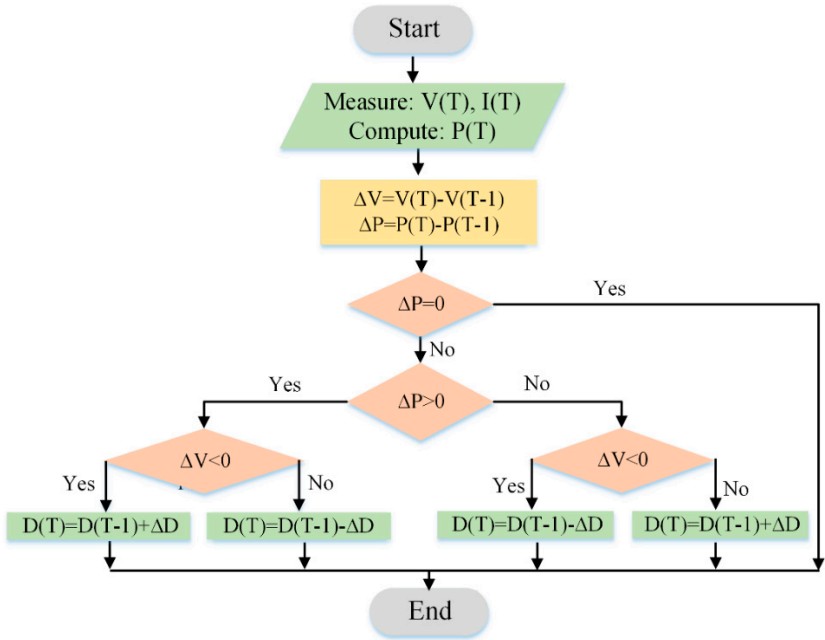

**Figure 4.** MPPT flowchart for the P&O algorithm.

**Table 1.** Perturbation summary for the P&O algorithm.

| Perturbation | ΔP | Resultant Perturbation |
|:---:|:---:|:---:|
| +ve | +ve | +ve |
| +ve | −ve | −ve |
| −ve | +ve | −ve |
| −ve | −ve | +ve |

*2.3. State of Charge (SOC) Estimation*

State of charge (SOC) is essential for LA batteries to declare their proper operation and to eliminate over-charging/deep-discharging issues. However, the nonlinear electrochemical responses mark the SOC estimation a challenging subject [25,26]. In the case of nonlinear systems, a BPNN is the most often employed technique because of its self-learning and feedback distinction [27]. Thus, a BPNN-based SOC assessment methods is adopted in this paper, as shown in Figure 5. As mentioned in Figure 6, SOC is estimated for every single battery independently by collecting its voltage and current parameters where $V_T$ is the total voltage of the battery bank. The root mean square error (RMSE) of the SOC estimation is minimized by adjusting the weights. In the BPNN, the total input of hidden layers can be calculated from:

$$neti_n = \sum_{i=1}^{3} x_i w_{in} + b_n \tag{5}$$

where $neti_n$ represents the total inputs to the hidden layer neuron $n$, $x_i$ symbolizes the value of input neuron, $w_{in}$ deliberates the weight between the input layer and hidden layer neurons $i$ and $n$; respectively, bias of the hidden layer neuron $n$ is mentioned as $b_n$.

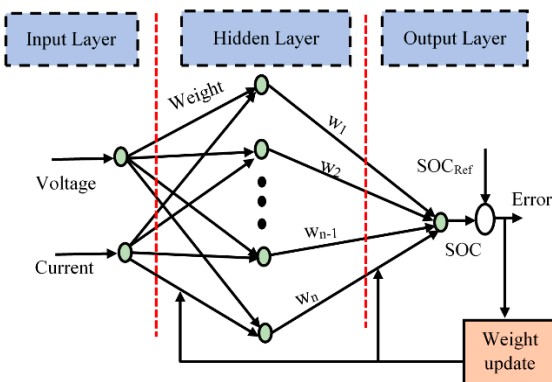

**Figure 5.** Structural design of the proposed BPNN algorithm.

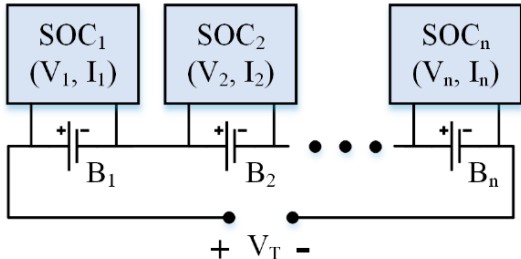

**Figure 6.** SOC estimation process from batteries.

A tangent function Equation (6) is the utilized activation function for the hidden layer neuron $n$. The valuation of the output layer neuron can be found from Equation (7):

$$h_n = f(neti_n) = \frac{1 - exp(-2neti_n)}{1 + exp(-2neti_n)} \tag{6}$$

$$neto = \sum_{i=1}^{n} h_i g_i + p \tag{7}$$

where, $neto$ represents the overall product of the output layer neuron $o$, the value of hidden layer neuron to output layer neuron for $i$ is $h_i$, the weight between hidden layer neuron $i$ and the output layer neuron $o$ is $g_i$, and the bias of the output layer neuron $o$ is $p$.

A sigmoid function is used as the applied activation function of the output layer neuron where output $y_n$ can be calculated from Equation (8):

$$y_n = f(neto) = \frac{1}{1 + exp(-neto)} \tag{8}$$

$$e_i = f_{Ref} - f_{Est} \tag{9}$$

$$\frac{\partial Error}{\partial w_{ij}^n} = \frac{\partial Error}{\partial f_{Est}} \frac{\partial f_{Est}}{\partial w_{ij}^n} \tag{10}$$

$$RMSE = \sqrt{(\sum_{i=1}^{n} e_i^2)/n} \tag{11}$$

where $e_i$ represents the output error, and $f_{Ref}$ and $f_{Est}$ are the reference output function and estimated output function, respectively. Using a gradient descent algorithm (GDA) as given by Equations (9)–(11), weights are restructured repeatedly to attain the expected output and reduce the error. From Equation (11) the RMSE is considered from which the improved weights are achieved while the error is diminished.

### 2.4. DC/DC Buck-Boost Converter

A circuit diagram of the used four switched synchronous DC/DC Buck-boost converter is shown in Figure 7 in which the battery cells are charged by the PV supply voltage $V_{PV}$ [28,29]. Four peculiar MOSFET switches ($S_1 \sim S_4$) are tangled into the DC/DC converter to standardize the energy transference of battery arrangements ($V_{nBat}$). The module can be operated in Buck mode, Buck-boost mode or boost mode by pursuing the energy requirement of the process. To activate the Buck mode operation, $S_3$ is turned off while $S_4$ is on. Additionally, $S_1$ and $S_2$ are activated to administer the procedure. MOSFETs $S_2$ and $S_1$ are opened and closed repeatedly to charge the inductor by the battery power. Inductor current increases while a capacitor supplies the output current for the charging duration. Furthermore, $S_1$ and $S_2$ are opened and closed frequently throughout the inductor discharging period to charge the load at this point. By altering the duty cycle to be d < 1, the average load voltage $V_{Load}$ (or $dV_{Bat}$) can be coupled together with the battery voltage $V_{Bat}$.

For the boost mode, MOSFETs S2 and S1 are turned off and on in a complement manner. S4 and S3 are opened and closed constantly by ensuring the inductor charging requirement. The arrangements are reversed at the discharging period where the average load voltage $V_{Load}$ can be calculated as $1/(1 - d)V_{Bat}$. However, the output load voltage at Buck-boost operation is $V_{Load}$ or $d/(1 - d)V_{Bat}$ which can be adjusted by following the desired battery voltage. MOSFET switches S2 and S4 are retained on to charge the inductor as well as S1 and S3 are preserved off for this mode. The MOSFET switches are activated on the inverse side to discharge the particular inductor. Table 2 summarizes the four switched synchronous DC/DC Buck-boost converter operation.

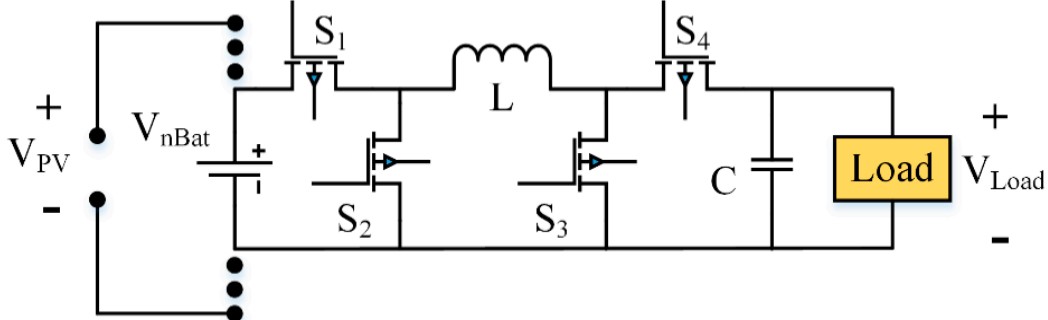

**Figure 7.** Four switched synchronous DC/DC Buck-boost converter.

**Table 2.** Summary of four switched DC/DC Buck-Boost converter operation.

| MOSFET Switch | Inductor Condition | Buck Mode | Boost Mode | Buck-Boost Mode |
|---|---|---|---|---|
| S1 | Charge | ON | ON | ON |
| | Discharge | OFF | OFF | ON |
| S2 | Charge | OFF | OFF | OFF |
| | Discharge | ON | ON | OFF |
| S3 | Charge | OFF | ON | ON |
| | Discharge | OFF | OFF | OFF |
| S4 | Charge | ON | OFF | OFF |
| | Discharge | ON | ON | ON |
| Average Load Voltage | $V_{Load} = dV_{Bat}$ | $V_{Load} = dV_{Bat}$ | $V_{Load} = \frac{dV_{Bat}}{1-d}$ | $V_{Load} = \frac{V_{Bat}}{1-d}$ |

*2.5. Battery Lifetime Estimation*

A Wöhler-curve-based aging model is considered in this research to estimate the battery lifetime [30,31]. This method is structured based on the battery DOD which is complementary to SOC. The DOD versus number of cycles for the Whöler curve of a lead-acid battery is shown in Figure 8. In this regard, the mathematical representation for battery lifetime loss $LL_i$ is expressed as:

$$LL_i = \frac{E_i}{E_i^{max}} \tag{12}$$

where $i$ indicates a certain DOD, $E_i$ is the number of events and $E_i^{max}$ is the maximum number of events which can be withstood for the battery. The entire loss of battery-lifetime $LL$ for the overall range of DOD (0~100%) can be calculated from:

$$LL = \sum_i LL_i \tag{13}$$

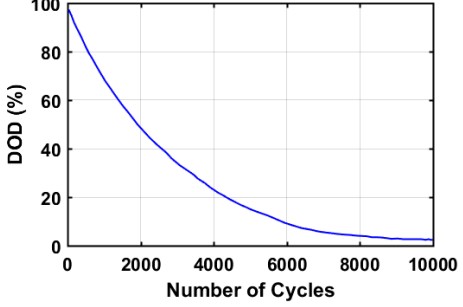

**Figure 8.** Wöhler curve for a lead-acid battery [30,31].

The battery is considered at its end of life when **LL** reaches 1. Thus, the total lifetime of the battery can be estimated from Equation (14) as the inversion of **LL** by considering the SOC profile.

$$Lifetime = \frac{1}{LL} \tag{14}$$

## 3. Proposed Methodology

### 3.1. Simulation Model

To validate the proposed BMS, the predicted model is simulated using MATLAB/SIMULINK 2019a software (MathWorks). The proposed methodology controls the MPPT scheme for PV panel and estimates the SOC of the battery bank through BMS in the designed model. The MPPT method generates the maximum possible power for the PV panel from the available solar insolation under any atmospheric conditions. On the other hand, the BMS receives the battery voltage and current from measuring sensors and estimates the existing SOC using BPNN algorithm. In this way, the charging and discharging process for the battery is performed through the PV supply and the connected load. As stated in [32–34], beyond 70~80% SOC the stocking battery exhibits over-charging, while lower than 30~40% SOC the holding battery runs in the deep-discharging manner. Thus, 40–80% SOC is ranged in this paper to perform smooth battery charging-discharging operation and, hence, lengthening the battery lifetime.

To standardize the battery bank capability and lessen the SOC imbalance among individual batteries, active battery balancing strategy is applied in the proposed model. As shown in Figure 9, the BMS generates a balancing control signal through the DC/DC Buck-boost converter by estimating the SOC from each individual battery. Every single battery SOC ($S_i$) is compared with the average SOC ($\overline{S}$). If the variance is in excess of a predefined threshold value ($S_{thr}$), the BMS instructs the relevant converter of a certain battery to carry on charging/discharging process according to Equation (15). In addition, a DC/AC power inverter is engaged to conform the validity of the proposed approach. The entire flowchart of the proposed BMS controlling technique is shown in Figure 10.

$$\begin{cases} S_i - \overline{S} > S_{thr} & Discharging \\ S_i - \overline{S} < S_{thr} & Charging \\ Others & Islanding \end{cases} \tag{15}$$

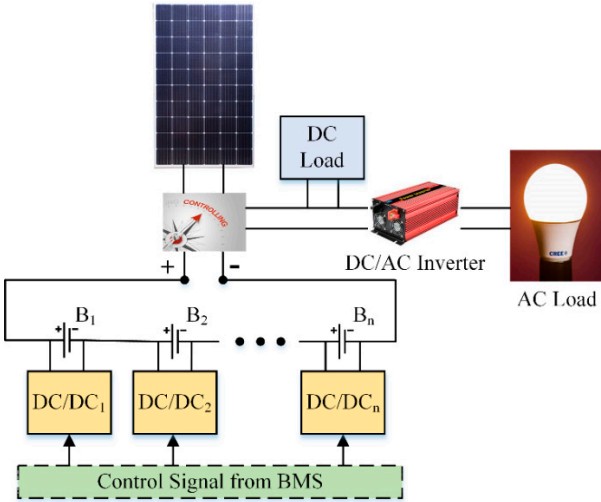

**Figure 9.** Active charge equalization structural design.

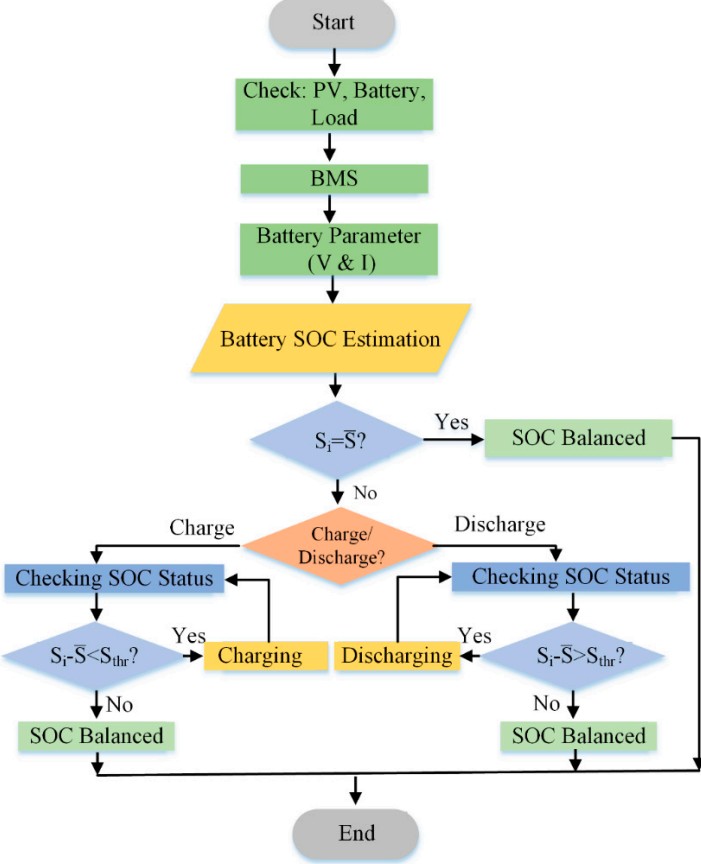

**Figure 10.** Flowchart for the proposed BMS.

*3.2. Experimental Model*

As shown in Figure 11, an experimental prototype of the projected model is prepared in a laboratory environment. The hardware prototype model consists of four 12 V, 7 Ah sealed gel rechargeable lead-acid batteries, solar module PV panel ($V_{OC}$ = 21.6V, $I_{SC}$ = 3.40A, $P_m$ = 50W) and 5W LED light as a load. In this prototype, the PV solar panel, battery, DC/AC inverter, and loads are connected through fuse and circuit breaker. MATLAB/SIMULINK 2019a software (MathWorks) is interfaced with an Arduino Integrated Development Environment (IDE) ATMega2560 controller. The integration of MATLAB Tools for Arduino IDE code generation block diagram is shown in Figure 12. Voltage and current sensors are used to calculate the voltage and current from the corresponding components and delivers the signal to a personal computer (PC) via the Arduino ATMega2560 electronic board. The SIMULINK model evaluates the data and appraises the PV power, battery SOC, and load power for the BMS controller. The DC/AC inverter converts the DC voltage into AC voltage which is formed as a modified AC sine wave. By imitating the effective condition of the governing mechanism, BMS directs the DC/DC Buck-boost converter to charge, discharge, or island the specific battery to balance the battery charge status and improve the battery-life proficiency.

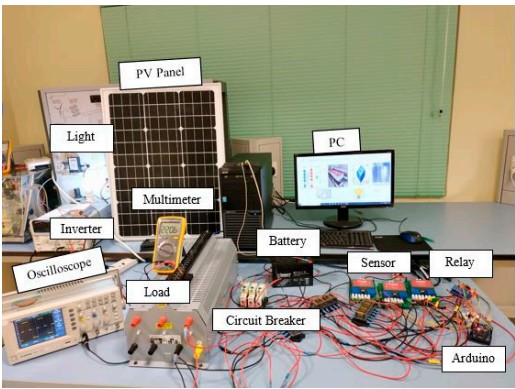

**Figure 11.** Experimental hardware setup of the proposed model.

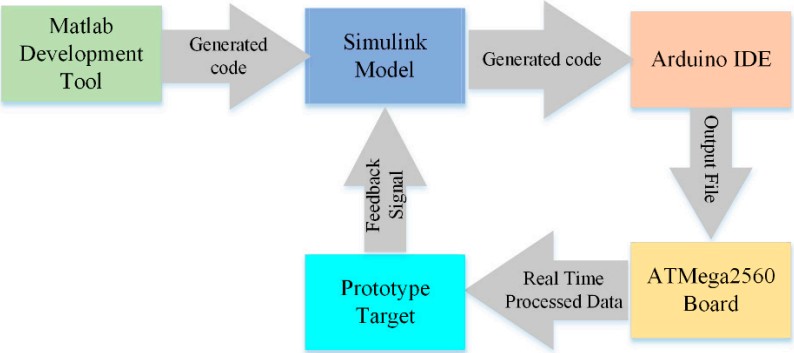

**Figure 12.** Integration of MATLAB-Arduino IDE code generation block diagram.

## 4. Results and Discussion

The experiment was conducted in January 2020 in the Renewable Energy Lab of Universiti Malaysia Sarawak (UNIMAS). Active battery balancing strategy accompanied by BMS for PV-Battery integrated system are conveyed together in the MATLAB-Arduino interfacing experience. Four series-connected lead-acid batteries are employed in this paper to validate the proposed battery balancing methodology for which the relationship between OCV vs. SOC is as shown in Figure 13 [35].

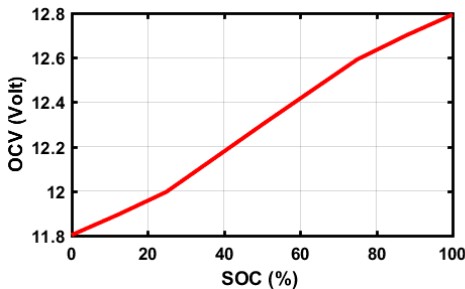

**Figure 13.** OCV vs. SOC for the employed 12 V lead-acid battery.

Initially, the batteries are comprised with dissimilar SOCs, such as 76%, 77%, 78%, and 79%. The PV supply voltage profiles at different atmospheric conditions (899~924 W/m$^2$ solar irradiance and 27~32 °C temperature) for the four series batteries are shown in Figure 14. Figures 15–18 demonstrate the battery balancing outcomes and exemplified some comparisons with and without the proposed strategy. Figure 15 compares the total battery bank voltage of the conventional and proposed method. It can be observed that the overall storage voltage for the proposed model is improved compared to the conventional method.

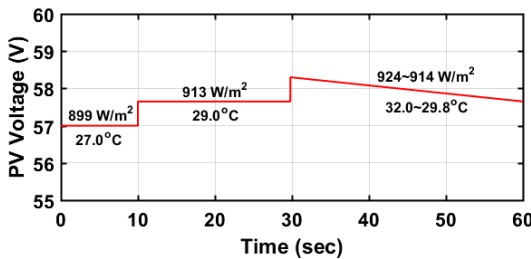

**Figure 14.** PV supply voltage at different atmospheric conditions.

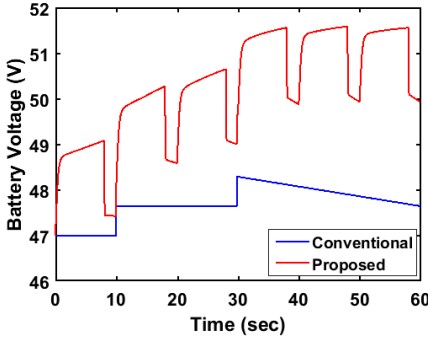

**Figure 15.** Overall battery bank voltage using conventional and the proposed method.

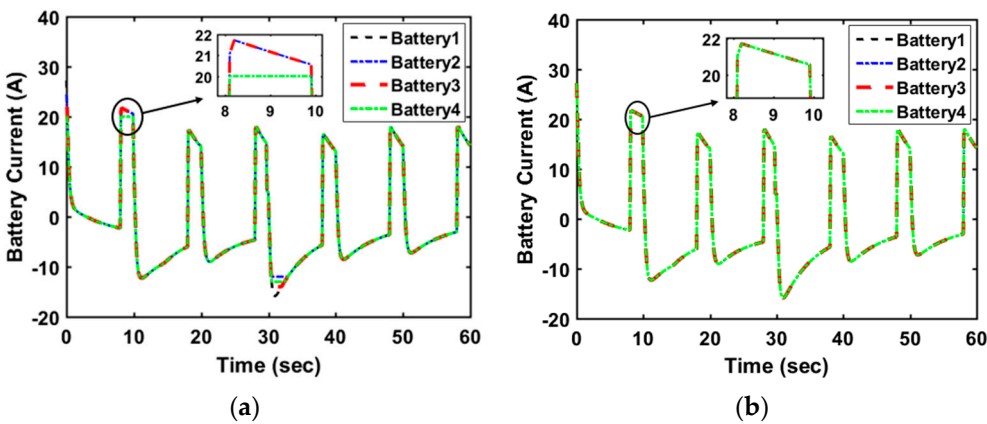

**Figure 16.** Current profile for individual batteries (**a**) without and (**b**) with the proposed method.

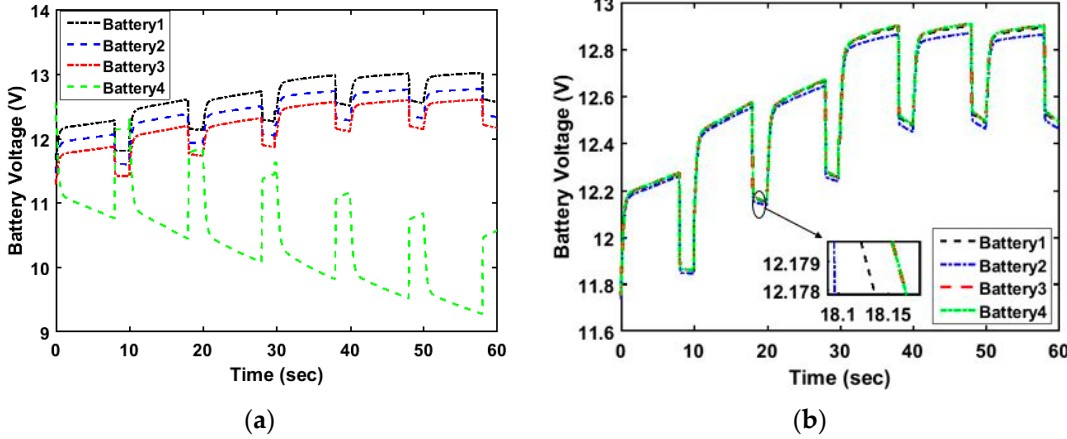

**Figure 17.** Individual battery Voltages (**a**) without and (**b**) with the proposed method.

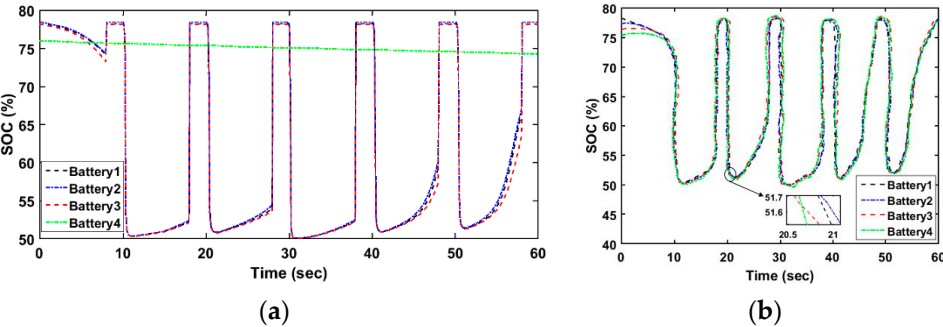

**Figure 18.** Individual batteries SOC (**a**) without and (**b**) with the proposed method.

Figure 16a reveals that although the supply current of the four series batteries should be the same, as a consequence of different SOC, the supply current is different without the implementation of any BMS scheme which may result in terminal voltage divergence among the batteries as can be noticed in Figure 17a. Implementation of the proposed BMS brings the cells current to nearly the same level as can be observed from Figure 16b which can limit the voltage differences among the four batteries as can be seen in Figure 17b. Figure 18 deliberates the SOC before and after the utilization of the proposed method. Without the proposed technique, the four batteries exhibit imbalance as can be shown in Figure 18a. With the utilization of the proposed methodology, the SOC of the four cells becomes close to 78.5% as shown in Figure 18b.

The fundamental proficiency of the battery energy storage system depends on the SOC estimation accuracy [22]. From Figure 19, SIMULINK toolbox trains the collected battery data in which the overall regression for training, testing, and validation data is about 97%. Therefore, the actual and estimated SOC is approximately the same as shown in Figure 20. SOC estimation accuracy (RMSE 0.082%) of the BPNN algorithm shown in Figure 21 and listed in the simulation results of Table 3 attest to the robustness of the proposed technique. The accuracy of the estimated SOC using the proposed technique outperforms the estimation accuracy using the invariant-imbedding method (IIM), Extended Kalman Filter (EKF), and Unscented Kalman filter (UKF) reported in the literature owing to the adopted weight observing scheme.

The produced DC voltage was transformed into a modified 240 V, 50 Hz AC waveform, as shown in Figure 22. A comparison between the simulation and experimental results of the obtained AC sine waveform is shown in Figure 23 which reveals good agreement of the experimental and simulation results. The key feature of the proposed BMS controller is the elongation of the battery storage functioning life. The misbalance of battery charging/discharging affects its performance and reduces its lifespan. Based on Wöhler-curve along with the proposed BMS, Figure 24 reveals that the projected model enhances the operational life of the battery compared to the conventional models.

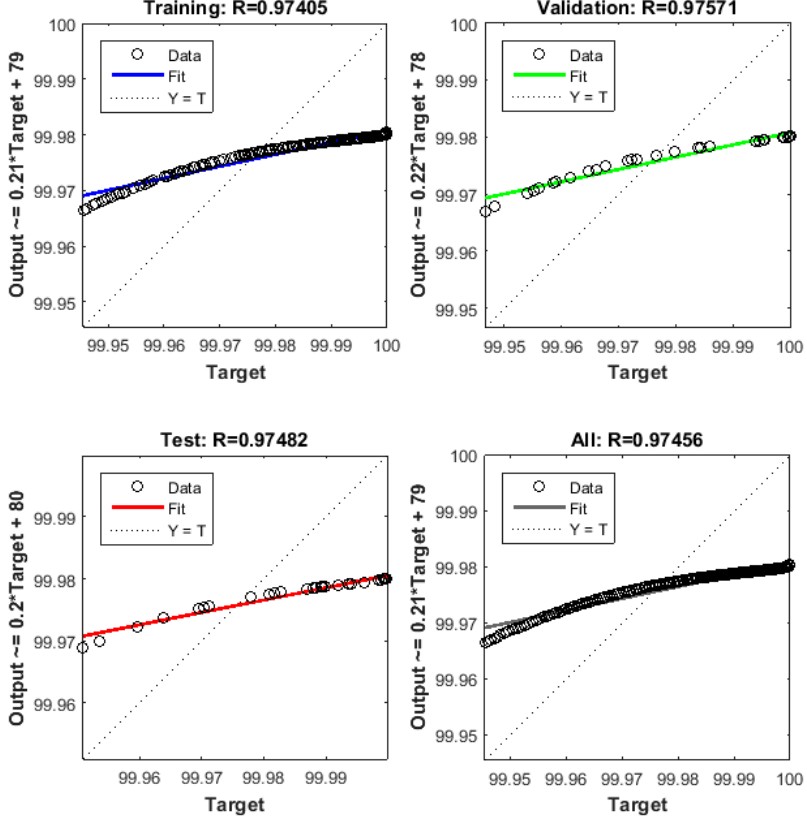

**Figure 19.** Training, testing, and validation results with regression.

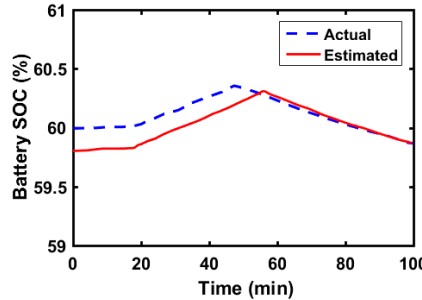

**Figure 20.** Comparison between actual and estimated SOC using the proposed technique.

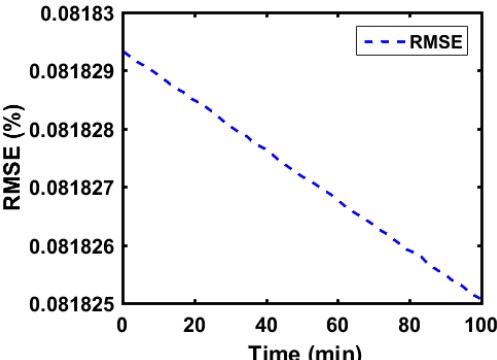

**Figure 21.** RMSE of the estimated SOC.

**Table 3.** SOC estimation accuracy comparison.

| Reference | Year | Battery Type | SOC Algorithm | Error (%) |
|-----------|------|--------------|---------------|-----------|
| [36] | 2016 | LIB | PI Observer | 3 |
| [37] | 2017 | LIB | EKF | 6.5 |
| [38] | 2017 | LIB | EKF | 3 |
| [39] | 2018 | LIB | UKF | 1.5 |
| [40] | 2019 | LIB | EKF | 2 |
| [41] | 2019 | LIB | BPNN | 1.2 |
| Proposed Model | 2020 | Lead Acid | BPNN | 0.082 |

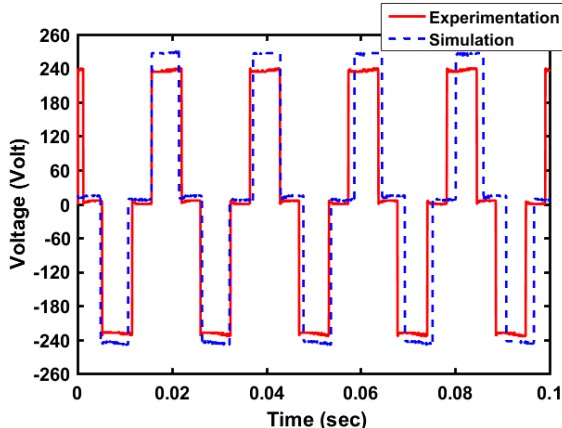

**Figure 22.** Experimental modified AC voltage.

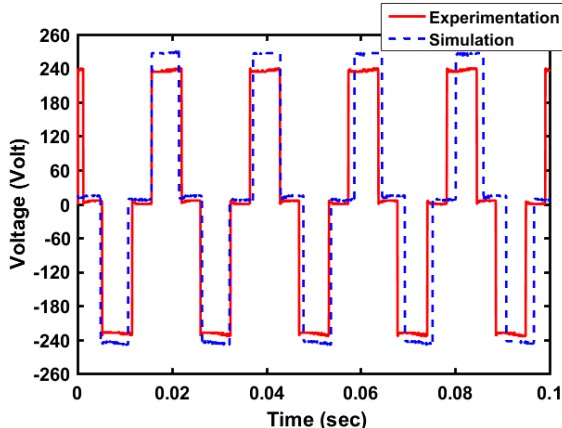

**Figure 23.** Comparison between simulation and experimental modified AC voltage.

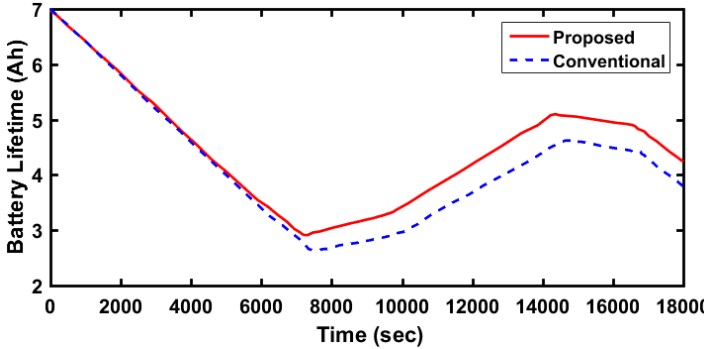

**Figure 24.** Battery lifetime comparison of the proposed and conventional models.

## 5. Conclusions

Battery storage is an essential, but expensive, element for PV-battery off-grid systems that ensures the continuity of the power supply at night and during cloudy days. This paper deliberates a new BMS which comprehends the PV model, P&O MPPT technique, and load and SOC estimation of battery storage systems. An active battery balancing approach based on adaptive BPNN and SOC estimation strategy has been proposed for to sustain the effectiveness of a battery pack system. The proposed model is applied on a series-connected battery string to eliminate the SOC disparity among individual batteries in the pack. The proposed BMS administers the SOC level of each battery and issues an activation signal to the corresponding DC/DC Buck-boost converter. Experimental analysis is conducted on the simulated model with the help of the MATLAB-ATmega2560 interfacing portion to validate the feasibility of the proposed technique. The proposed control strategy detects the charge status of the lead-acid battery and takes decisions to accomplish the expected operational goals and elongate the battery lifetime with 0.082% RMSE in the estimated SOC. Further research is recommended to replicate this study for lithium-ion batteries as well as on-grid storage systems.

**Author Contributions:** M.O.Q. conceived the original idea; M.O.Q. and Y.B. wrote and edited the manuscript; and M.L.H. and A.A.-S. supervised the study and revised the final version of the paper. All authors have read and agreed to the published version of the manuscript.

**Funding:** This research was funded by Research and Innovation Management Center (RIMC), UNIMAS via Fundamental Research Grant Scheme, Ministry of Higher Education, Malaysia, grant number FRGS/1/2017/TK10/UNIMAS/03/1.

**Conflicts of Interest:** The authors declare no conflict of interest.

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
