# Peer review of "Active Charge Balancing Strategy Using the State of Charge Estimation Technique for a PV-Battery Hybrid System"

_energies, doi:10.3390/en13133434_

Round 1
Reviewer 1 Report
The main remarks are:
- This algorithm is related to lead acid batteries; however, lithium batteries present the new trend in BESS. Which seem to be the main challenges if you apply your proposed methodology to this battery type?
- Lines 69-73: It is essential to provide more details for the novelties of your paper. Also, similar additional information can be also provided in Section 3, where you analyze your methodology.
- Does the used PV model in Matlab/Simulink contains info which is related to realistic PV operation, such as effect of dust, partial shading, hotspots, PID effect, etc?
- In general, more details have to be given for the developed Matlab/Simulink models, regarding PV, batteries, etc.
- In the experimental model, the batteries you are using are deep-cycle batteries?
- Table 3: Apart your proposed methodology, all other results refer to simulation models?
- Image resolution in Fig. 4 and Fig. 9 has to be improved.
Author Response
Pleas have a look to the attached file

Reviewer 2 Report
Please see the attached comment sheets.

Author Response
Please have a look to the attached file.

Reviewer 3 Report
This paper proposes a method for SOC estimation and battery balancing. This topic is very important to enhance the performance of BMS. The method has also well described with validation. The paper is generally well written. It can be accepted after address the following issues:
- The state estimation is one major contribution of this paper, but has not been discussed in the literature review. Some discussions should be added for SOC estimation like some recently published works, eg. Applied Energy 268 (2020): 114932; IEEE Transactions on Industrial Electronics, DOI: 10.1109/TIE.2019.2962429; IEEE Transactions on Industrial Electronics 66.7 (2018): 5724-5735.
- Please explain more on the results in fig 23. How is it obtained?
- Advantages and disadvantage of the proposed technique in this field should be better discussed for justification.
Author Response

(The authors gave the same response as above.)

Round 2
Reviewer 1 Report
All remarks have been answered
Reviewer 2 Report
All comments are cleared.